# Tuning the reactivity of carbon surfaces with oxygen-containing functional groups

Jiahua Zhou[1,2,6], Piaoping Yang[1,2,6], Pavel A. Kots [1], Maximilian Cohen[1], Ying Chen [3], Caitlin M. Quinn [4], Matheus Dorneles de Mello [2,5], J. Anibal Boscoboinik[2,5], Wendy J. Shaw [3], Stavros Caratzoulas [2], Weiqing Zheng [2] ✉ & Dionisios G. Vlachos [1,2] ✉

Oxygen-containing carbons are promising supports and metal-free catalysts for many reactions. However, distinguishing the role of various oxygen functional groups and quantifying and tuning each functionality is still difficult. Here we investigate the role of Brønsted acidic oxygen-containing functional groups by synthesizing a diverse library of materials. By combining acid-catalyzed elimination probe chemistry, comprehensive surface characterizations, [15]N isotopically labeled acetonitrile adsorption coupled with magic-angle spinning nuclear magnetic resonance, machine learning, and density-functional theory calculations, we demonstrate that phenolic is the main acid site in gas-phase chemistries and unexpectedly carboxylic groups are much less acidic than phenolic groups in the graphitized mesoporous carbon due to electron density delocalization induced by the aromatic rings of graphitic carbon. The methodology can identify acidic sites in oxygenated carbon materials in solid acid catalyst-driven chemistry.

Owing to their low cost, light-weight, high surface area, high thermal conductivity, and readily modified structure and surface chemistries, carbon-based catalysts are routinely used for energy storage, sensors, electrocatalysis, and heterogeneous catalysis[1–3]. Pure carbon with a uniform $sp^2$-hybridized structure is not suitable for catalysis. Functionalization with heteroatoms[4,5], including N, O, P, etc., makes carbon a prominent support and a metal-free catalyst. The microenvironment of carbon atoms[4,6–8] strongly impacts the interaction between reactants, intermediates, and products with catalysts[9,10]. Undoubtedly, oxygen-containing functional groups (OCFGs) are widely investigated in many applications[10–12]. It has been suggested that the composition in acidic OCFGs (−OH, −COOH, lactone, carboxylic anhydrides) affects the metal particle size distribution by serving as coordination sites for metal cations, greatly dispersing metal particles[13]. Furthermore, several OCFGs can serve as Brønsted acid sites (BAS), e.g., for dehydration, cellulose hydrolysis,

etc[14–16]., or base sites. The relative activity of each type of site is unknown.

Developing OCFGs composition-function relationships remains elusive due to having many different sites and limitations on characterization stemming from challenges in using ultraviolet and visible light-based spectroscopic techniques on carbon[17]. Specifically, Fourier transform infrared spectroscopy (FTIR), often used to distinguish sites based on the vibrational frequency of the chemical bonds, is inapplicable, as IR photons are almost completely absorbed by carbons. While diffuse reflectance IR spectroscopy (DRIFTS) can be applied, obtaining IR extinction coefficients is challenging[18,19]. Temperature-programmed decomposition mass spectrometry (TPDE-MS) can detect OCFGs over a temperature range (500 to 1200 K). Yet, surface functionalities may undergo consecutive transformations giving gas-phase products formed in situ during TPDE rather than being in the initial carbon structure[20–22]. Raman spectroscopy is promising to

[1]Department of Chemical and Biomolecular Engineering, University of Delaware, Newark, DE 19716, USA. [2]Catalysis Center for Energy Innovation, University of Delaware, Newark, DE 19716, USA. [3]Pacific Northwest National Laboratory, Richland, WA 99352, USA. [4]Department of Chemistry and Biochemistry, University of Delaware, Newark, DE 19716, USA. [5]Center for Functional Nanomaterials, Brookhaven National Laboratory, Upton, NY 11973, USA. [6]These authors contributed equally: Jiahua Zhou, Piaoping Yang. ✉e-mail: weiqing@udel.edu; vlachos@udel.edu

detect the crystal and molecular structure and very sensitive to the structural disorder of carbon but cannot distinguish OCFGs[23]. Boehm titration[24,25] assumes bases of different strength react with a specific type of OCFGs but is challenged when distinct chemical functions possess similar pKa values. Furthermore, it cannot distinguish aprotic acidic groups (lactones, carboxylic anhydrides) that can hydrolyze into −OH or −COOH in water or acid/base solutions and does not account for the porosity and material hydrophilicity/hydrophobicity (in the liquid phase)[16,17,25]. High-resolution X-ray photoelectron spectroscopy (XPS) is sensitive to the elemental composition and the relative carbon fraction in various oxidation states. Blume et al.[26,27] reported high-quality fits of the C 1s spectra. However, some binding energy signals unavoidably overlap partially due to similar oxygen chemical states in various OCFGs within only a 5 eV energy range. For example, lactones and carboxylic groups containing O = C−O cannot be unambiguously identified through peak fitting. The −OH can be hydroxyl (associated with alkyl rings) or phenolic (associated with benzene rings), but it is challenging to distinguish the two. Therefore, it is essential to develop general techniques to understand the role of oxygen functionalities in such complex materials as carbons.

Here we investigate the chemical functionality of carbons by synthesizing a diverse library of materials with varying oxygen functionalities. We combine multimodal techniques, namely ex situ and in situ XPS and TPDE-MS, gas-phase elimination probe reactions (*tert*-butanol and isopropanol dehydration)[28], and machine learning (ML) tools (regression, partial least squares, etc.)[29,30] to elucidate the protonated acidic OCFGs (−OH, −COOH) serving as Brønsted acid sites. Unexpectedly, we find that −OH (phenolic group) is the dominant site

and extremely more active than −COOH (carboxylic group). By introducing [15]N isotopically labeled acetonitrile adsorbed magic-angle spinning nuclear magnetic resonance (MAS NMR) as a general acid site identification method for carbons, and comprehensive density functional theory (DFT) calculations, we demonstrate that the higher activity of −OH stems from the carbon microenvironment and specifically the electron density delocalization caused by aromaticity and the number of benzene rings conjugated with the conjugate base of the acid site.

## Results

### Identification and quantification of oxygen-containing functional groups (OCFGs)

Graphitized mesoporous carbon (GMC) was chosen on account of its highly graphitic composition (Supplementary Fig. S1), low fraction of surface functional groups, heteroatoms, and limited density of imperfections, such as twists, non-aromatics, links, and vacancies. Nitric acid oxidation involves oxidative ions (e.g., $H_3O^+$, $NO_3^-$, $NO_2^+$, arising from self-dehydration) that attack the aromatic rings, change the conjugate carbon atoms (Fig. 1a), and introduce vacancies, defects, and edge planes. Collectively, nitric acid oxidation forms basic and acidic groups (carbonyls, carboxylic, phenolic groups, etc.)[31]. The oxidized materials were subsequently annealed and are hereafter denoted as GMCs-ox (a h)-b °C (acid treatment for hours and annealed at temperature b; see Methods). Figure 1b shows that the intensity ratio of the D-band to G-band ($I_D/I_G$) of the oxidized GMCs-ox is lower than the as-received-GMCs, revealing conversion of the graphitized carbon to defects during treatment (conditions in Supplementary Table 1).

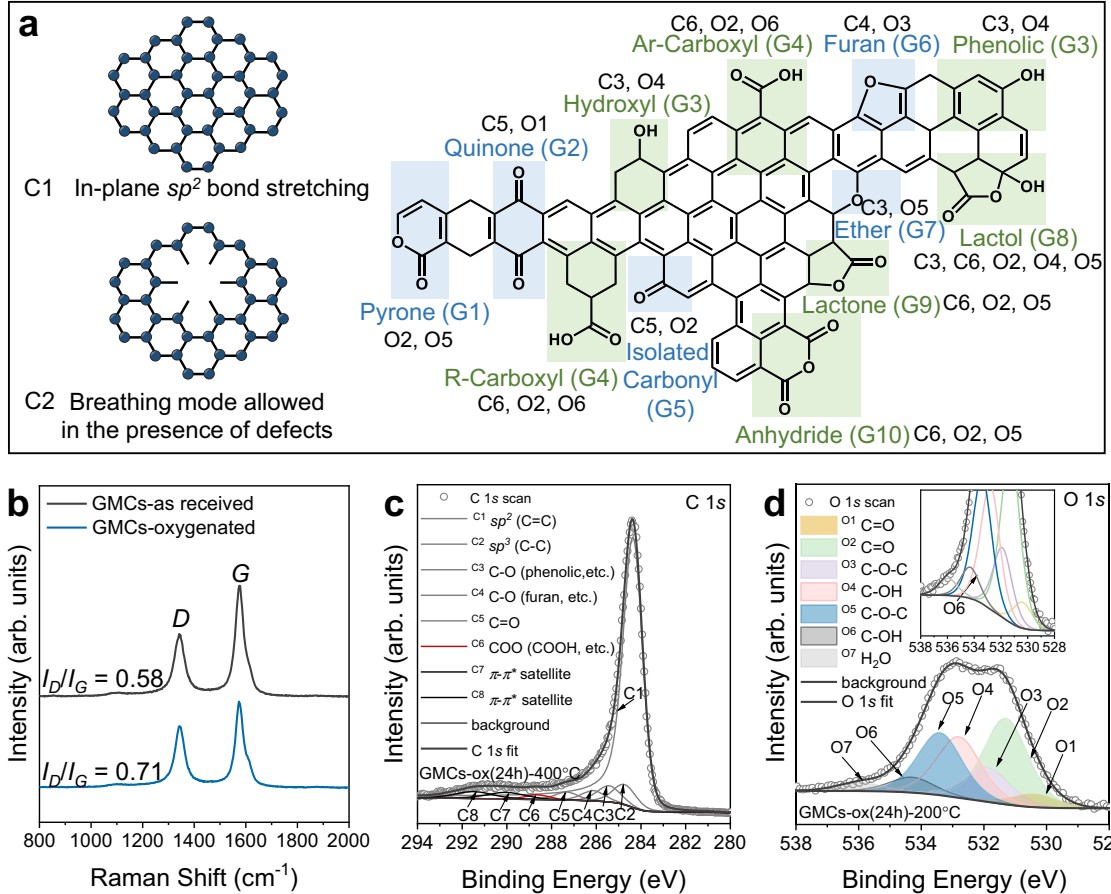

**Fig. 1 | Carbon surface microenvironment. a** Graphitic model structures and oxygen-containing functional groups (OCFGs): green and blue shadings indicate acidic and basic OCFGs, respectively. **b** Raman spectra of as-received GMC and GMC-ox (oxygen-functionalized) normalized to the D-band intensity of the GMC-as

received sample. **c** C 1s peak fitting of GMCs-ox (24 h)−400 °C and **d** O 1s peak fitting for GMCs-ox (24 h)−200 °C (inset enlarged view of Fig. 1d) for showcasing the approach.

Both XPS analysis and CHNS measurement (Supplementary Table 1) indicate the introduction of oxygen into the carbon without other heteroatom contamination e.g., nitrogen, (Supplementary Fig. 1a, Table 2, Fig. 3). The elemental composition can be determined from the C 1s and O 1s core-level spectra with sensitivity factors from the Kratos library[20] (Supplementary Table 1). Figure 1c, d exhibits the representative and stable C 1s and O 1s fitting models[26,32] of two samples, with detailed fitting basis and assignments, fitting parameters, and Monte Carlo error analysis in Supplementary Note 1. The quantification of the species from the C 1s spectra (Supplementary Fig. 4a, c; Fig. 5) is disturbed by the asymmetric structure of the graphitic carbon peak (group C1, $I_{max} = 284.4$ eV) up to high binding energies. Differences in OCFGs (range higher than 285 eV) can still be observed by subtracting the GMCs-ox C 1s spectra from the as-received GMCs (Supplementary Fig. S4). Minor errors in the subtraction of the graphitic peak result in a large error due to the small OCFGs fractions, highlighting one of the challenges in data analysis[33]. Although assigning the chemical bonds contributing to the O 1s core-level spectrum (Fig. 1d) is less straightforward[34–36], the differences in spectra among samples are again statistically significant (Supplementary Fig. 3b, d). Below, we use spectra differences as part of our approach. The various carbon Ci (species from C 1s fitting) and oxygen Oi (species from O 1s fitting) groups depicted in Fig. 1a that are quantified from the XPS data, e.g., C3, C6, or O4, do not in most cases correspond to a single functional group and thus to a single active site, but rather to a collection of a few of them. Conversely, individual oxygen species may appear in several XPS groups, correlating XPS groups. These correlations complicate the data analysis and development of structure-reactivity relations. For example, standard principal component analysis of the groups affecting the reaction rate gives unphysical results. It is thus crucial to account for these correlations to reveal the actual active sites.

To unequivocally unravel the distribution of groups, we combine XPS with TPDE-MS (Fig. 2a) of the CO, $CO_2$, $H_2O$ released upon heating the samples (previously annealed at various temperatures). Comparison of the pristine and annealed samples reveals important mechanistic information on the stability of the various groups. As shown in Fig. 2a, the broad $CO_2$ peak in the pristine GMCs-ox (48 h) sample from 150 to 400 °C with a maximum around 250 °C is attributed to the decomposition of carboxylic groups (group O6)[37]. The shoulder at 400–600 °C stems from the decomposition of carboxylic anhydride and lactones (groups in O2 and O5), which are more stable than the carboxylic groups (group O6). The $H_2O$ peak at 250 °C is attributed to the dehydration of phenolic/hydroxyl groups (group O4) or the formation of carboxylic anhydrides from the condensation of neighboring carboxylic groups (group O6). Notably, the significant peak at 600–650 °C in the CO profile arises predominantly from the degradation of phenolic or ethers from the C−O bond (groups in O3, O4, O5; Fig. 1a). Peaks higher than 650 °C are ascribed to C=O from ether, ketone, or quinone groups (groups in O1, O2, O5).

Figure 2b displays the fittings of the O 1s difference spectra of samples annealed at various temperatures. The data are correlated with the $H_2O$, $CO_2$, and CO MS profiles. The difference spectra between GMCs-ox (48 h)−200 °C and GMCs-ox (48 h)−400 °C can be fitted with four peaks (Fig. 2b, bottom). The 531.2 eV is assigned to the C=O double bond of (group O2) ketones, lactones, anhydrides, etc.; the 531.9 eV to the C−O−C in furans or keto-enolic tautomers of group O3; the 532.7 eV to the phenolic group (group O4), and the 534.2 eV to the carboxylic group (group O6). Consistent with the TPDE profile of the pristine GMCs-ox (48 h) and GMCs-ox (48 h)−400 °C, the carboxylic group (group O6) is mainly removed, and the phenolic group (group O4) is partially condensed upon annealing at 400 °C. In addition, the phenolic/hydroxyl species (group O4) in the difference spectra of GMCs-ox (48 h)−400 °C and GMCs-ox (48 h)−600 °C (Fig. 2b, middle) are largely decomposed upon annealing at 600 °C. In contrast,

carboxylic groups (group O6) almost fully disappear, in line with the MS profiles. The protonated groups O4 and O6 are completely eliminated when annealed at 800 °C (Fig. 2b, top) due to their lower thermal stability than species in group O2 and O3. These findings agree with MS data. In summary, we conclude that the MS data and the fittings of the O 1s difference spectra furnish comprehensive data for quantifying the OCFGs from the XPS fitting model we use in the present work. Figure 2c, d shows the distribution of $sp^2$ vs. $sp^3$-hybridized carbon and OCFGs in the oxidation and annealing steps and provides mechanistic understanding during the treatments. Notably, after six hours of nitric acid treatment, the graphitic ($sp^2$-hybridized) carbon has decreased significantly, indicating structure damage at early times, giving rise to $sp^3$ carbon[20]. At longer oxidation times, the $sp^2$ and $sp^3$ carbons change more gradually. The oxygenated groups more likely form early on at the more reactive $sp^3$ carbons of pentagons or heptagons. Consistent with the MS profiles, the density of protonated O4 and O6 groups is anticorrelated with the annealing temperature but positively correlated with the oxidation time. Clearly, the acidic groups (phenolic, carboxylic, anhydride, lactone, etc.) are predominantly obtained by low-temperature liquid-phase oxidation. In contrast, the basic groups (ketones, quinones, and isolated carbonyl groups) form primarily upon annealing in an inert gas at temperatures > 400 °C. These expected findings can guide future materials synthesis. Overall, various oxygenated groups are simultaneously introduced upon acid treatment and further adjusted with thermal treatment.

## Correlating alcohol dehydration with multiple-acid active centers

Expectedly, the acidic and basic OCFGs give the material acid-base bifunctional properties[37,38]. The base properties result probably from the adsorbed $O^-$ and $O_2^-$ species or pyrone-like structures[39], whereas the acid properties stem from protonated species including carboxylic (−COOH) and phenolic/hydroxyl groups (−OH). Other OCFGs with complicated structures, such as lactones, are non-catalytic. Water or an acidic/basic environment can hydrolyze some of these OCFGs into protonated groups (for example, carboxylic anhydrides form two carboxylic groups and lactones generate one carboxylic and a neighboring phenolic group, respectively)[16]. This interconversion of groups, potentially during reaction, can complicate the development of structure-reactivity relations as the number and type of sites may occur at time scales comparable to those of chemical reactions (see below).

To reveal the Brønsted acidic nature of OCFGs, we employed tert-butanol (t-BuOH) dehydration to isobutene as a probe acid-catalyzed reaction[28,40,41]. Figure 3a depicts time-on-stream (TOS) data for the as-received and the four oxidized carbons, illustrating the initial rate (circles on the y-axis). The rate varies slowly with TOS. The practically zero dehydration rate over the as-received GMC (Fig. 3a, bottom) indicates minimal acidity. In stark contrast, samples with abundant acidic OCFGs (Fig. 2c) exhibit high initial dehydration rate (Fig. 3a, middle and top). However, the initial dehydration rate does not scale linearly with the oxygen content (Fig. 3b). Specifically, samples with similar oxygen content can show profoundly different initial rates by an order of magnitude (see data with O between 1 and 4 mmol/$g_{cat}$; gray ellipse). A similar rate is obtained using materials with an oxygen content varying significantly from 1 to 3 mmol/$g_{cat}$. (Bottom data in Fig. 3b; blue ellipse). Consequently, the variation in type, strength, and carbon hybridization makes it challenging to correlate the Brønsted acid active sites to the dehydration rate.

It is challenging to unravel the contribution of the dominant OCFGs (individual groups, Fig. 1a, Fig. 3g) to the alcohol dehydration (Fig. 3a, b) due to the overlap of binding energies among functional groups in the XPS spectra. The C 1s and O 1s core-level XPS spectra fitting provides information about multiple groups at once, as shown in Fig. 1a. We employ stoichiometric relationships between the XPS

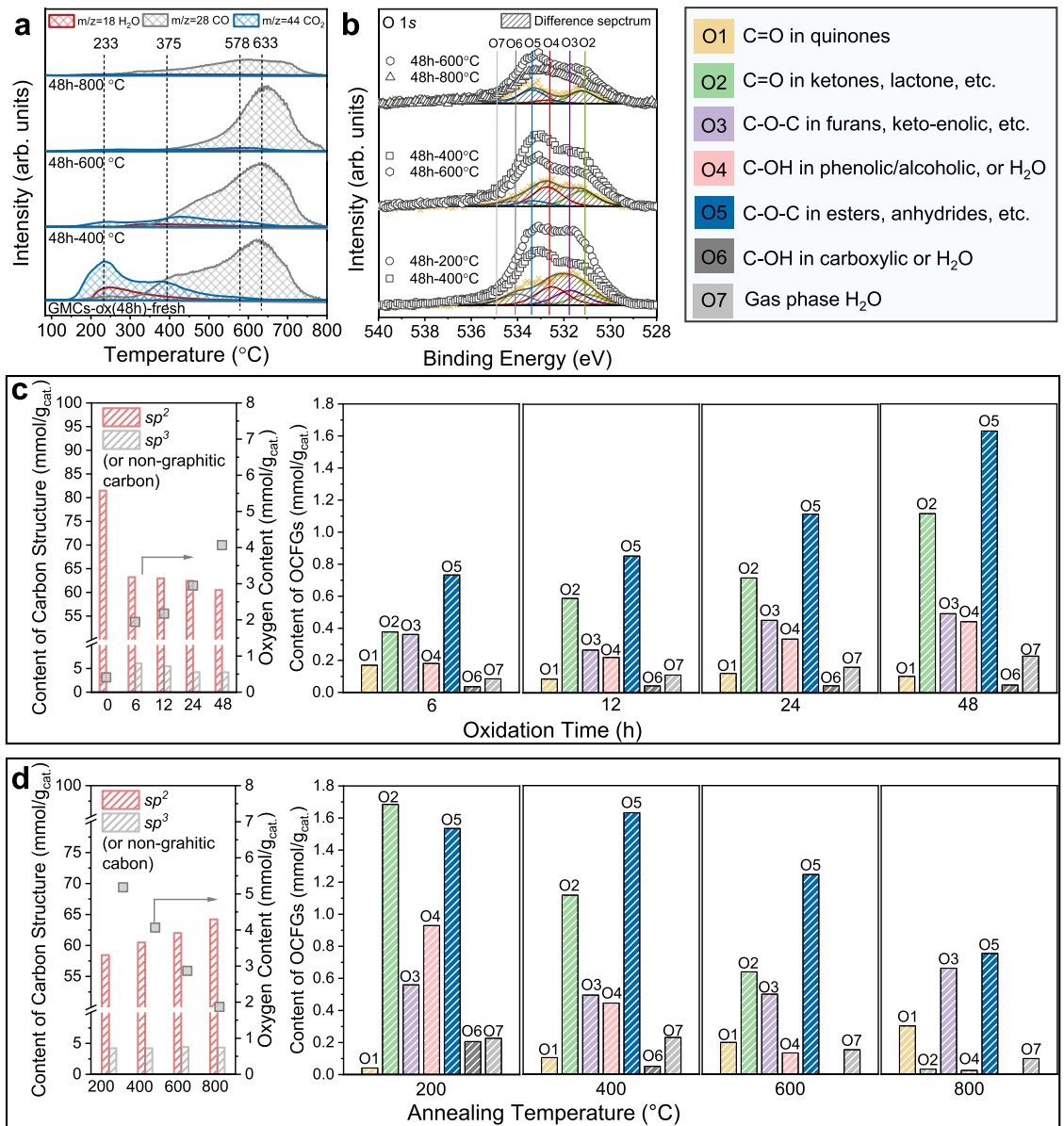

**Fig. 2 | Carbon hybridization and OCFGs at various conditions. a** TPDE-MS profiles of CO, $CO_2$, and $H_2O$ for GMCs-ox (48 h)-fresh, GMCs-ox (48 h)-b °C, (b = 400, 600, and 800 °C). **b** O 1s ranges of XPS spectra and difference spectra (hatched areas including the fit). Concentration of carbon and functional groups from C 1s and O 1s fittings during pretreatment at **c** various oxidation times, and **d** annealing temperatures after oxidation for 48 h.

fitting groups of the fresh sample (**C3-C6** ($j = 1–4$) and **O1-O6** ($j = 5–10$)) and the intrinsic oxygen functional groups (**Gi** ($i = 1–10$)) to quantify the latter and build OCFGs-reactivity relations (See Supplementary Note 3). Before decoupling the OCFGs into individual groups (**Gi**), we correct the surface oxygen content obtained from XPS with bulk content measured by CHNS (Supplementary Note 2). We then investigate the OCFGs-reactivity relationship using elementary machine learning tools including correlation matrix, ordinary least squares regression, and partial least regression. To eliminate the dynamic change of the OCFGs during reaction, we consider the initial reaction rate and the group concentration obtained from the fresh samples for these data analyses. The correlation matrix indicates that three acid groups, namely **G3** (phenolic/hydroxyl), **G4** (carboxylic), and **G10** (carboxylic anhydride), positively impact the dehydration activity. **G3** exhibits the strongest positive correlation (0.93) with the dehydration rate (Fig. 3c). In contrast, the basic groups, except for **G5** (isolated carbonyl), correlate weakly with the reaction rate. The strong positive effect of **G5** on the reaction rate could merely be the result of the

correlation between **G3** and **G5**. Likewise, the ordinary least squares regression of the dehydration rate with **Gi** (Fig. 3d) also suggests that **G3** is the dominant active site, further confirmed by the partial least regression (Supplementary Fig. S8). The 95% credible intervals of the **G3** (hydroxyl/phenolic) site, showing the lowest root-mean-squared error (RMSE) range (Bayesian analysis, Supplementary Fig. S9) without overlapping with any alternative candidates and the best linear regression statistically, deduce it as the optimal active site.

To better understand the dynamics of surface species upon thermal treatment and exposure to $H_2O$, we employ 2-propanol (IPA) as the probe molecule and the typical acid-catalyzed dehydration[37] to propylene on acid sites and dehydrogenation to acetone catalyzed by acid-base pairs (Fig. 3e). At 250 °C, over the oxidized GMCs, 2-propanol undergoes almost exclusively dehydration to propylene and non-dehydrogenation product (acetone) was detected, indicating a surface free of Lewis acid[42], acid-base pairs and metal sites[43,44]. As shown in Fig. 3e, the steamed catalyst is less active than the fresh one but more active than the annealed catalyst, suggesting that the fast deactivation

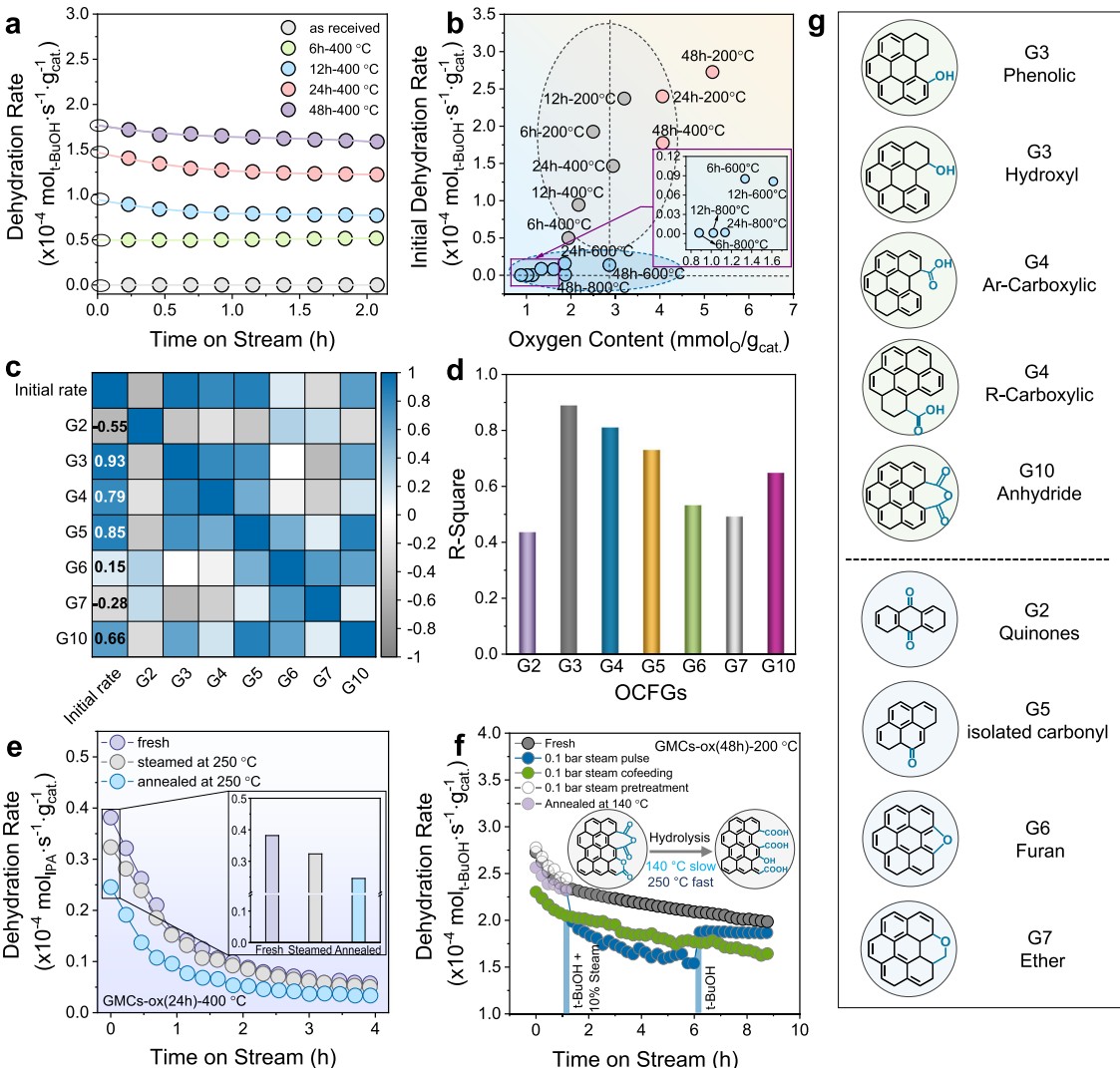

**Fig. 3 | Probe reaction and key reactivity descriptors. a** *tert*-butanol dehydration rate on oxidized samples. Reaction conditions: 140 °C, 4 mol% t-BuOH, 10 mg catalyst. The initial rates are estimated using a 3rd-order polynomial fit and extrapolated to zero reaction time. **b** Correlation between oxygen content and initial rate. **c** Correlation matrix, showing the correlation coefficients between descriptor pairs. The purple and gray colors indicate a strong positive and negative correlation between factors. **d** R square of scatter plot of the initial rate against the concentration of OCFGs. **e** Dynamics of active sites with pretreatment conditions (inset initial rate with treatment conditions). Pretreatment conditions: annealing for 1 h; steaming for 1 h with 0.1 bar steaming; carrier gas of 100 mL/min $N_2$/He or $N_2$/water mixture. Reaction conditions: 250 °C, 4 mol% 2-propanol, 100 mL/min $N_2$/2-propanol mixture, 50 mg catalyst. **f** Dynamics of active sites during reaction at 140 °C (inset: scheme of hydrolysis of non-protonated acidic groups). Reaction conditions: 140 °C, 4 mol% t-BuOH, 10 mg catalyst, carrier gas 100 mL/min water/$N_2$ mixture. **g** Structure models of individual groups: green and blue shadings indicate acidic and basic OCFGs, respectively.

is due to the loss of active sites (thermally desorbed and coking) (in situ XPS, Supplementary Figs. 10–12; ex situ XPS, Supplementary Figs. 14, 15; TPDE-MS, Supplementary Fig. 16); and the in situ formation of protonated carboxylic group (**G4**) (from hydrolysis of carboxylic anhydrides by steaming in 0.1 bar $H_2O$ at 250 °C) compensates for the deactivation resulting from thermal decomposition (air free XPS of steamed sample, Supplementary Fig. 17). To eliminate the complication from the in situ formed −COOH (from hydrolysis of carboxylic anhydride) causing difficulties in obtaining intrinsic activity of active species, we investigate the dynamics of active sites with various reaction conditions in t-BuOH dehydration (Fig. 3f). Water co-feed from the beginning (Fig. 3f) gives intuitively a decreased initial dehydration rate due to enhancement of the reverse, equilibrium-limited dehydration reaction[28]. Water co-feed in the middle (Fig. 3f) of the reaction slightly dropped the reactivity compared with the one cofeeding water from the beginning, suggesting the minor compensating effect of water. 1 h steaming and annealing pretreatments (Fig. 3f) over the catalyst at

140 °C influence the dehydration rate slightly, suggesting minor and very slow hydrolysis of anhydride at low reaction temperature (140 °C). Therefore, we exclude the unprotonated acidic groups (**G10**, carboxylic anhydride) from the active sites contributing to the initial dehydration reactivity at 140 °C.

With the number of active centers (**G3**, −OH; **G4**, −COOH) quantified and the exclusion of the **G10** as possible active center, we construct a two-parameter kinetic model and three one-parameter models to estimate the TOF of t-BuOH dehydration on each active site. (See details in Supplementary Note 4.) Due to the dynamic change of the OCFGs during reaction (Supplementary Figs. 14, 15), we correlate the overall t-BuOH initial dehydration rate with the number of −OH and/or −COOH in the four models (Supplementary Table 20). The two-parameter kinetic model (model A) estimated that the TOF of −OH (phenolic/hydroxyl) groups is higher than that of −COOH (Supplementary Table 20); notably, the TOF of −COOH could be lower, even zero, due to the confidence interval including zero (Supplementary

Table 21, model A), further suggesting that −OH (phenolic/hydroxyl) groups are more active than −COOH groups. Furthermore, model C, which assumes −OH is the unique active site, exhibits the best regression, further indicating −OH is more active than −COOH for alcohol dehydration. This is a counterintuitive result; aliphatic alcohols are known to be less acidic than their corresponding acids. We examine the physical basis and the dehydration mechanism using calculations and introduce NMR for acid site characterization of carbons next.

**Mechanistic study and acid site verification**

To understand the reasons behind the higher activity of −OH, we performed DFT calculations for the dehydration of 2-propanol over −OH (phenolic) and −COOH associated with benzene rings. We demonstrated that most of the −OH were in the form of phenolics (−OH associated with benzene rings) and −COOH are showed as the Ar−COOH (−COOH associated with benzene rings) due to the non-reduction preparation and treatment atmosphere. The active sites, −OH and −COOH, were grafted onto terminal sites of the armchair and zigzag graphene ribbons; the optimized geometries of the four models are shown in Fig. S18. The Brønsted acid-catalyzed dehydration of IPA is well understood, and our calculations did not reveal any deviations from the E1 mechanism (Fig. 4a, b). After binding, the IPA hydroxyl accepts a proton from the Brønsted site (−OH or −COOH). A water molecule is eliminated, and the resulting intermediate carbocation is stabilized by the conjugate base (−O$^-$ or −COO$^-$). We considered the ensuing β-elimination to propylene both in the presence and absence of the produced water molecule at the catalyst surface. The free energy profiles for the overall reaction on the two sites are compared in Fig. 4c. Two observations are in order: (a) dehydration on the −OH site requires 0.3 eV less activation energy than on the −COOH site, revealing that the former is a stronger Brønsted acid, in agreement with our experimental data; (b) the β-elimination is slightly more facile on the −O$^-$ group than on the −COO$^-$ group by 0.12 eV in the presence of the water molecule but the order is reversed when the water molecule is not at the surface. The latter has no kinetic implications, however, because clearly the water elimination (first step) is rate limiting. Additionally, we note that both intermediates (AH_C$_3$H$_8$O* and A_C$_3$H$_7$_H$_2$O*) in the first step are less stable on the −OH site than on the −COOH site but the stability of transition states (TS1) exhibits an opposite trend. This can be understood by the differences in hydrogen bonding on the two sites. Specifically, IPA bound to the −COOH site participates in two hydrogen bonds, −H$_2$O$_2$···O$_1$H$_1$− and −H$_1$O$_1$···O$_3$− (Supplementary Fig. 20a) but only in one hydrogen bond, −H$_2$O$_2$···O$_1$H$_1$−, when it is bound to the −OH site (Supplementary Fig. 20b). This explains the difference in stability of the intermediates prior to reaction. The corresponding transition states, however, are stabilized by only one hydrogen bond each (O$_2$-O$_1$, Supplementary Fig. 20c, d). On the −OH site, the H-bond is stronger than on the −COOH site as the respective O$_2$-O$_1$ H-bond lengths are 2.534 and 2.646 Å. After protonation and C−O bond cleavage, which proceed in a concerted manner (Supplementary Fig. 20e, f), the O$_2$-O$_1$ bond on the −OH site is longer than on the −COOH site (2.863 Å vs. 2.798 Å), leading to a less stable intermediate.

At first sight, the conclusion that we have a stronger Brønsted acid in −OH than in −COOH might be surprising. However, consideration of the bond structure around the grafting sites can readily explain it. The conjugate base of −COOH prefers to delocalize its excess charge over its two oxo groups. On the other hand, the conjugate base of −OH can delocalize electron density over a network of conjugated double bonds furnished by the aromatic rings of the support and, as a result, is more stabilized, resulting in the hydroxyl's stronger acidic character. That this is indeed the case can be seen in Fig. 4d, where we plot the excess electron densities of the two conjugate bases. It is evident that −O$^-$ delocalizes its electron density over the carbon atoms of the

neighboring aromatic rings of the support more extensively than −COO$^-$. Thus, the relative acidity of the −OH and −COOH groups is essentially determined by the support. The deprotonation energies (DPE) of the two sites seem to be aligned with these observations as the DPE of −OH is smaller than that of −COOH by 0.16 eV. The effect of delocalization on the Brønsted acidic character of the hydroxyl is also evident in the varying DPE with the number of the fused aromatic rings in the following pairs: phenol, benzoic acid; 1-naphthol, 1-naphthoic acid; 1-anthrol, 1-anthracenecarboxylic acid; 1-pyrenol, 1-pyrenecarboxylic acid; and 1-coronenol, coronene-1-carboxylic acid. Figure. S21 in the Supporting Information illustrates the transition from phenol being less acidic than benzoic acid to 1-coronenol being more acidic than coronene-1-carboxylic acid.

To assess the DFT calculations and provide further evidence of the acid sites, we have used $^{15}$N MAS NMR of absorbed acetonitrile (ACN, a weak base) from the gas or liquid phase to distinguish the acidity strength of −OH (phenolic/hydroxyl) and −COOH (Ar/R-carboxylic) due to the high sensitivity of magnetic shielding to the local electronic environment of the observed nucleus. $^{15}$N MAS NMR spectra of ACN chemisorption from the gas phase at very low surface coverage show peaks of ACN strongly bonded to acid sites (Supplementary Fig. 22). The chemical shift of the acidic OCFGs is lower than the as-received GMCs. Figure 4f depicts the correlation between the $^{15}$N chemical shift and the −OH/−COOH concentration ratio based on XPS analysis. Notably, the chemical shift of $\delta_{iso}$ ($^{15}$N) decreases with increasing −OH/−COOH ratio (quantified by XPS), suggesting the average −OH in this carbon material has stronger Brønsted acidity than the average −COOH. Due to the high mobility of ACN, even at low coverage, and acid site heterogeneity, the experimentally measured value of $\delta_{iso}$ ($^{15}$N) is effectively a superposition of ACN at different binding locations (−OH and −COOH). To confirm this, ACN was deposited from liquid phase at surface coverage above monolayer with gradual desorption ($^{15}$N NMR result, Supplementary Note 5, Figs. 23, 24). $^{15}$N MAS NMR reveals four different peaks. Based on the relative desorption rate, the peaks were assigned to liquid phase ACN, strongly/loosely physically absorbed ACN, ACN absorbed on weak acid sites, and ACN absorbed on strong acid sites. However, it is impossible to distinguish the exact position of −OH and −COOH, even in spectra measured at low temperatures (−25 to −75 °C), due to the complex carbon microenvironment around acid sites. It was reported that the main interaction of the probe molecule ($^{15}$N ACN) with the hydroxy (in zeolite) is via a N···H−O type hydrogen bond to the Brønsted proton. Until recently, the isotropic shift of nitrilic nitrogen atoms ($\delta_{iso}$ $^{15}$N) was believed to be a descriptor of the BAS acidity strength; $^{15}$N chemical shifts to a lower frequency (decrease in ppm) correspond to strong BAS[45–47]. The chemical shift can be reflected in the N···H bond length; a stronger N···H bond (i.e., a shorter the r (N···H)) shields nitrogen more, and shifts the electron density toward the nitrogen atom, resulting in the $^{15}$N chemical shift to lower frequency[45]. Consistently, the theoretical chemical shift of $^{15}$N over −OH is 260.43 ppm, which is smaller than on −COOH (264.79 ppm). The calculated N···H bond length shown in Fig. 4g further verifies the lower chemical shift of −OH than −COOH. Thus, $^{15}$N NMR of absorbed ACN can be effectively applied to estimate the average surface acidity of −COOH and −OH, despite ambiguities concerning the exact position of each site type (phenolic −OH or Ar/R−COOH), which is further proved by ACN absorbed at low and high coverages.

## Discussion

Oxygenated carbon materials as metal-free catalysts have been demonstrated in many vital applications, but the catalytic activities of the functional groups have remained elusive. The combination of XPS and TPDE-MS, computational modeling, machine learning (data analysis) and the dehydration probe reactions can reveal the acidic OCFGs. Our study further demonstrates that the acidic groups

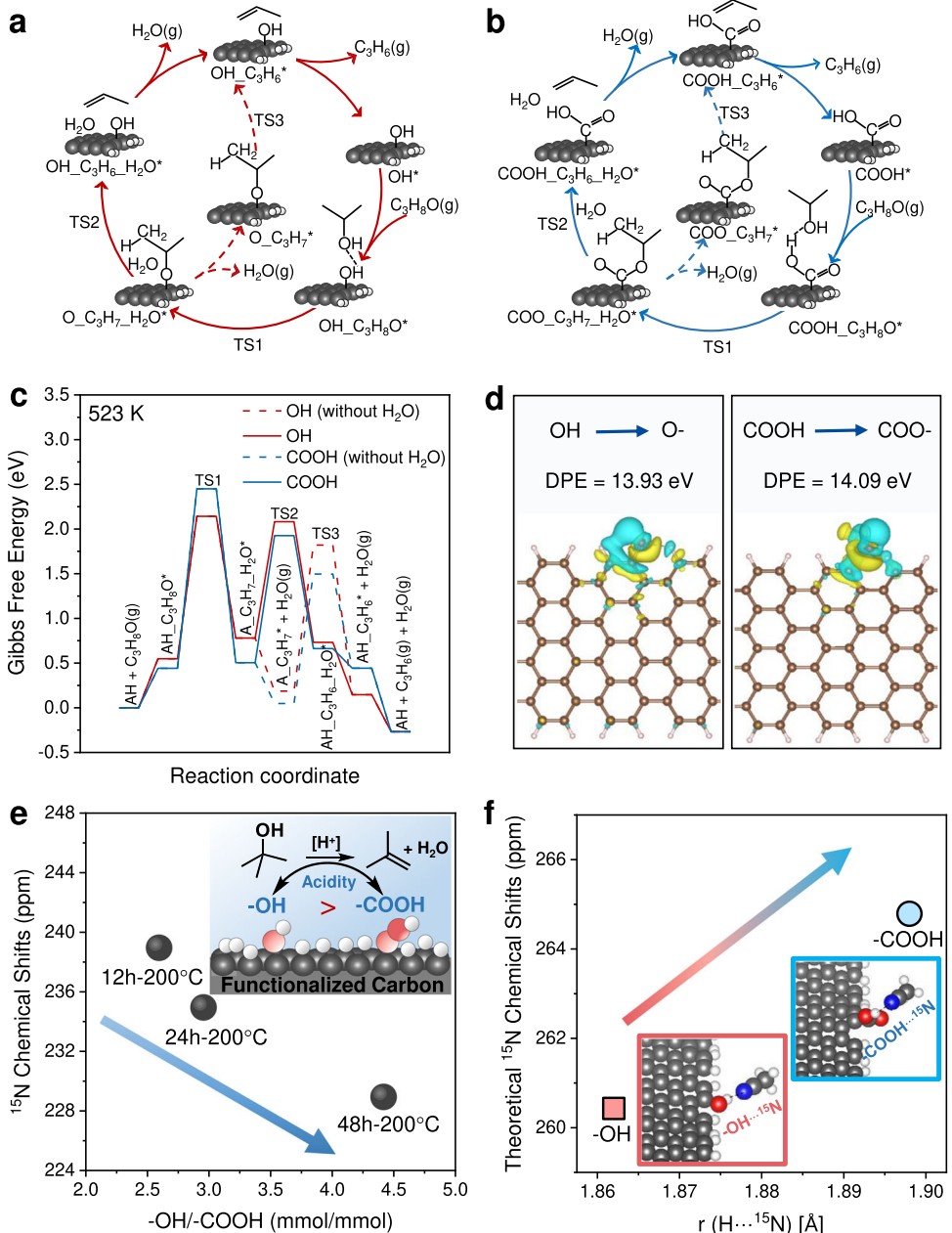

**Fig. 4 | Reaction pathways of IPA dehydration on the on acidic OCFGs and strength of acidity.** IPA dehydration pathways at **a** −OH and **b** −COOH. **c** Corresponding Gibbs free energy profiles at 523 K (Brønsted acid site and conjugate base denoted by AH and A, respectively). **d** Delocalization of conjugate base excess charge. **e** ¹⁵N chemical shifts (from gas-phase absorbed ¹⁵N ACN) against −OH/−COOH concentration ratio. **f** Theoretical isotropic ¹⁵N chemical shifts of C≡¹⁵N···H for −OH and −COOH vs. H···N bond length.

(hydroxyl/phenolic, Ar/R-carboxylic, anhydride groups) can significantly be modulated by changing the pretreatment conditions (thermal treatment, steaming at various temperatures). The non-protonated acidic anhydride can be hydrolyzed to form in situ protonated groups in gas-phase reactions at high steaming temperatures (250 °C) but slower at low temperatures (140 °C). The −OH (phenolic/hydroxyl) and −COOH (Ar/R-carboxylic) are the dominant acidic active sites. Strikingly, the higher activity of −OH (phenolic) than −COOH (Ar-carboxylic) is evidenced by DFT calculations, NMR characterization, and deprotonation energies. While OCFGs are simply classified as acidic and basic, the reactivity of OCFGs can counterintuitively change profoundly by the carbon microenvironment because of the electron density delocalization resulting from the neighboring aromatic rings or the various OCFGs. Using probe

reactions, we have developed a methodology for analyzing complex systems. Combining probe reactions and multimodal characterizations, we identified the acidic groups and mathematically quantified the active species on carbon surfaces. The methodology for site characterization can be applied to more complex reactions that involve carbon as a support. Overall, the analytical approaches and findings presented here can be extended to other carbon-based acid site-driven heterogeneous catalysis and materials containing other heteroatoms.

## Methods

### Preparation of functionalized carbon materials

Graphitized mesoporous carbon (GMC) (Sigma-Aldrich, <500 nm, >99% trace metal basis) was first treated with concentrated nitric acid

(Sigma-Aldrich, 70%, purified by redistillation, ≥ 99.999% trace metals basis) under vigorous magnetic stirring at 130 °C for varying time. The ratio of nitric acid to carbon was set as 150 mL to 6 g to keep the concentration of $HNO_3$ constant. After treatment, the solution was cooled to room temperature, centrifugated, and washed with deionized water until the pH reaches 7. Then the carbon samples were dried at 110 °C for 24 h. Finally, the samples were annealed in a tube furnace at different temperatures for 1 h under Helium with a 100 mL/min flow rate and a heating ramping of 1.5 °C/min. The samples are labeled as GMCs-ox (a h)-b °C, with a and b corresponding to the $HNO_3$ treatment time (a = 6 h, 12 h, 24 h, 48 h) and annealing temperature (b = 200 °C, 400 °C, 600 °C, 800 °C), respectively.

## Raman characterization
The Raman spectra were taken on the HORIBA Raman Spectrometers and Microscopes equipped with a green line solid-state laser with 532 nm laser source. The N.D. filter fixed at 5% of the laser power. The spectra were acquired on different spots and then an average was taken. The Raman spectra analysis was performed using the LabSpace 6 Spectroscopy Software.

## Specific surface area (BET) measurement
The textual properties of the oxygenated carbon samples were measured by $N_2$-adsorption using a Micrometritics Tristar II 3000 Analyzer at 77 K. The pore size distribution was determined by the Barrett–Joyner–Halenda (BJH) method, and the specific surface area was calculated from the isotherms using the Brunauer–Emmett–Teller (BET) method.

## Temperature-programmed decomposition (TPDE-MS)
The TPDE-MS experiments of the oxidized GMCs materials were carried out in an inhouse-built fixed bed reactor operating at near atmospheric pressure in helium. Typically, 50 mg of the catalyst with a particle size range of 40-60 mesh was loaded into a quartz tube and dehydrated at 80 °C for 30 min in the flow of 50 mL/min He. Subsequently, the sample was heated to 800 °C (10 °C/min), during which effluent was monitored using the mass spectrometry (MS, Omnistar, Pfeiffer). The following AMUs were used for analysis: $H_2O$ ($m/z$ = 18), $CO_2$ ($m/z$ = 44), CO ($m/z$ = 28), and He ($m/z$ = 4).

## Elemental analysis
Elemental analyses (CHNS) were performed with an Elementar Vario Cube CHNS, equipped with the thermal conductivity detector, and the C, H, S traps for separation in the Advanced Materials Characterizations Lab at the University of Delaware. The sample combustion temperature can be heated up to 1200 °C. The precision is ~1% for homogeneous samples.

## X-ray photoelectron spectroscopy
Ex situ XPS measurements were conducted in a thermo-fisher K$\alpha$ (1486.6 eV) + X-ray photoelectron spectrometer equipped with a monochromatic aluminum K$\alpha$ X-ray source (300−400 nm). The scanning parameters are 0.1 eV for the survey and detailed scanning. The powdered catalyst sample was pressed on a Cu foil to prevent contamination from other carbon resources.

The near ambient pressure XPS (NAP-XPS) experiments were conducted at the Center for Functional Nanomaterials (CFN), Brookhaven National Laboratory (BNL). The carbon materials were scanned with a lab-based SPECS Surface Nano Analysis spectrometer. The instrument was equipped with a SPECS Phoibos analyzer and a monochromated Al (1486.6 eV) photon source. The sample powders were pressed to a cleaned thin Cu foil and transferred into the vacuum chamber. C 1s and O 1s survey spectra were collected during all procedures. The oxidized GMCs samples in UHV were treated in the preparation chamber of the XPS setup at various temperatures. For the

dehydration reaction used in the in situ XPS studies, the samples were exposed to 1 torr isopropyl alcohol (IPA) at 250 °C for 100 min.

All XPS spectra were processed using the CASA XPS version 2.3.22 PR1.0 by using a Shirley background, and the C 1s features were calibrated based on the $sp^2$ peak at 284.4 eV referenced to the highly oriented pyrolytic graphite (HOPG) peak and fitted with an asymmetric line shape with the Doniach−Sunjic (DS line-shape) model[48]. The rest of OCFGs in the C 1s region were constrained with symmetric line-shape (Gaussian−Lorentzian mixture) and the same full-width half-maximum (FWHM) based on the structure and graphitization of carbon materials[26,27] (detailed fitting procedure is summarized in Supplementary Note 1). To obtain high-quality fits, we adjusted the relative sensitivity factors (RSF) in the component until the effective RSF was equal to unity. The spectra were fitted with a mixed Lorentzian-Gaussian function.

The fraction of carbon atoms with $sp^2$-hybridization (%) was calculated based on XPS analysis (Eq. 1)

$$Csp^2(\%) = \frac{\%_C \times \%_{sp^2_{\text{area ratio from C 1s fitting}}}}{100} \qquad (1)$$

Here $\%_C$ is the percent of the atomic carbon and $\%_{sp^2}$ area ratio from C 1s fitting is the percent of the peak area of $sp^2$ from the C 1s deconvolution result.

The fraction of OCFGs (%) was calculated from the XPS data (Eq. 2)

$$OCFGs(\%) = \frac{\%_{C(\text{or } O)} \times \%_{OCFGs_{\text{area ratio from C 1s or O 1s fitting}}}}{100} \qquad (2)$$

Here $\%_O$ is the percent of oxygen and $\%_{OCFGs}$ area ratio from C 1s or O 1s fitting is the percent of the peak area of the oxygen functional groups from C 1s and O 1s deconvolution.

The determination of $x_j$ (molar concentration of OCFGs types normalized to catalyst weight, $mol_C/g_{cat}$ or $mol_O/g_{cat}$) from the XPS peak ratios is based on (Eq. 3):

$$\begin{aligned} x_j &= XPS_{\text{peakratio}} / \text{carbon(or oxygen) molar mass} \\ &\quad \times \text{carbon (or oxygen) mass weight} \\ &= \frac{mol\, C_{\text{typej}}}{mol\, C_{\text{total}}} \times \frac{mol\, C_{\text{total}}}{g\, C_{\text{total}}} \times \frac{g\, C_{\text{total}}}{g_{\text{cat}}} \end{aligned} \qquad (3)$$

To convert atomic percent from XPS to weight percent, we use (Eq. 4):

$$O\%(\text{weight percent}) = \frac{O\%(\text{atomic percent}) \times 16}{(1-O\%) \times 12 + O\% \times 16} \qquad (4)$$

The quantification of C, O content was obtained from the core-level C 1s and O 1s spectra, with the relative sensitivity factors C 1s 0.296 and O 1s 0.711, respectively. The calculations are based on the following equation (Eq. 5)

$$\frac{n_i}{n_j} = \frac{I_i/S_i}{I_j/S_j} \qquad (5)$$

The calculation of the OCFGs concentration is mainly based on the assumption that oxygen is distributed uniformly throughout the spherical carbon.

## $^{15}N$ solid-state MAS NMR spectroscopy
Gas-phase absorption procedure: prior to the adsorption of the probe molecule ($^{15}N$ labeled acetonitrile, ACN, Cambridge Isotope Laboratories, Andover, MA), a desirable amount of catalyst was first placed in a fixed-bed reactor at near ambient pressure. The sample was first degassed at 100 °C in argon (Ar, 99.999% purity) for 1 h. The

samples then were treated with the mixture of $^{15}N$ labeled acetonitrile and Ar was injected by a syringe pump, and the flow rate was controlled at 50 mL/min. The treatment was performed at room temperature (RT, 25 °C) and then completed when the concentration of $^{15}N$ ACN reached an equilibrium (~60 min). The tail gas in the whole process was analyzed online continuously by gas chromatography (GC, Agilent 7890 A) equipped with a flame ionization detector. The sealed reactor was transferred into an $N_2$-filled glovebox. The treated sample was then packed and sealed in a 3.2-mm zirconia NMR rotor inside the glove box. $^{15}N$ and $^1H$ one-dimensional spectra were collected on a 14.1T Bruker Avance III spectrometer with a 3.2 mm HCN probe, spinning at 12–14 kHz MAS. The sample temperature was maintained at 25 °C. All spectra were acquired with $^{15}N$ direct polarization, a 5.1 μs 90° pulse, and a 5 s recycle delay. 5120–15360 scans were acquired per sample. For $^1H$ NMR spectra, a 180°–90° sequence with a 10 ms interpulse delay was used to reduce the background signal. All $^1H$ spectra were acquired with a 3.4 μs 90° pulse, 512 scans, and a 3 s recycle delay.

Liquid-phase absorption procedure: 20 μL of $^{15}N$-enriched ACN was added to 80 mg of carbon materials, mixed, then sealed for 1 h at room temperature. The mixture was open to glove box at room temperature for 5, 15, and 60 min before packed into NMR rotors for $^{15}N$ ACN evaporation. Single-pulse $^{15}N$ MAS NMR experiments of liquid-phase absorbed ACN were performed on a Varian-DDR 20.0T NMR spectrometer (with a $^{15}N$ Larmor frequency of 86.12 MHz) using a commercial 3.2 mm pencil-type probe with a rotor spinning rate of 10–20 kHz. The typical parameters for acquiring quantitative $^{15}N$ spectra were: pulse width = 2 μs with a tip angle of 30 degree, recycle delay time = 10 s (an array of recycle delay varying from 5 s to 60 s confirmed that 10 s is sufficient for reaching equilibrium state between each scan), spectrum width = 34.7 kHz, acquisition time = 100 ms, number of scans = 1200–4000, and temperature range = −100 °C to 25 °C (low temperature was controlled using a liquid nitrogen tank).

### Probe reactions

The probe reactions were carried out in a quartz reactor with an internal diameter of 9 mm. The catalyst bed was fixed by two inert quartz beds (30-40 mesh) and was placed in the isothermal zone of a furnace. A thermocouple, covered by a thin quartz tube, was placed in the middle of the reactor to infer the catalyst bed temperature. Mass flow controllers (Brooks) controlled the flow rates; Nitrogen ($N_2$) (>99.999%) was used as the carrier gas for purging tert-butanol into the reactor, and helium (He) (>99.999%) served as the balance gas. The total flow rate of the gas mixture was 100 mL/min.

The probe reactant was introduced to the reactor through a gas saturator with controlled temperature. The alcohol flow rate was adjusted through the flow rates of $N_2$ and He and the bubbler's temperature. All pipelines were heated to >150 °C by heating tapes to prevent alcohol condensation. The reaction products were analyzed using gas chromatography (Agilent 7890 A) equipped with a Plot-Q column and a flame ionization detector (FID). The catalyst was pretreated in a mixture of $N_2$ (50 mL/min) and He (50 mL/min) at 140 °C for 1 h before purging alcohol into the reactor. The initial dehydration rate was calculated by extrapolating the data to zero time (Eq. 6).

$$Dehydration\ rate = \frac{n_{\text{initial}_{t-BuOH}} \times Yield_{\text{iso-butene}}}{time \times m_{\text{cat}}} \quad (6)$$

### Machine learning (ML)

Non-negative Least square regression (NNLS) and partial least squares (PLS) were applied to simplify the complexity of the high-dimensionality data and relate the multiple types of OCFGs with the dehydration rate. These various elementary machine learning were carried out using Python software.

### DFT calculations

Spin-polarized DFT calculations were performed using the Vienna ab initio software package (VASP, version 5.4.1)[49]. The electron exchange and correlation effects were described by Perdew–Burke–Erzenhof (PBE) exchange-correlation functional[50]. The core electrons were represented with the projector augmented wave (PAW)[51] method and a plane-wave cutoff of 500 eV was used for the valence electrons. Van der Waals interactions were considered via dispersion-corrected density functional theory calculation (DFT-D3)[52] in the simulations. The Gaussian smearing method with a smearing width of 0.05 eV was employed. The pristine graphene ribbon with armchair edges contained 54 C atoms and 12 H atoms, while the one with zigzag edges consisted of 60 C atoms and 12 H atoms. The vacuum space in the z-direction was set to 25 Å and the in-plane vacuum between ribbons was set to at least 15 Å. Brillouin zone was sampled with a $(1 \times 3 \times 1)$ k-point and $(3 \times 1 \times 1)$ k-point grid for the armchair edge model and the zigzag edge model, respectively. All geometry optimizations were performed using the conjugate gradient algorithm. The atomic force convergence of 0.02 eV/Å and the energy tolerance of $10^{-6}$ eV were employed. The total energies of the gases were calculated in boxes of $20\,Å \times 21\,Å \times 22\,Å$ using gamma point. To obtain thermal corrections of electronic energies at 523 K, the vibrational frequencies were computed within the harmonic oscillator approximation via diagonalization of the Hessian matrix using the central difference approximation with a displacement of 0.015 Å. Transition states were computed using nudged elastic band and dimer calculations[53] and confirmed by vibrational frequency calculations. Thermochemical parameters of gaseous species were taken from the Burcat database[54]. The free energies of surface species were corrected using the python Multiscale Thermodynamic Toolbox (pMuTT)[55].

The deprotonation energy (DPE) was defined as the energy required to separate the proton from the acid group, as shown below (Eq. 7):

$$DPE = E_{H^+} + E_{A^-} - E_{AH} \quad (7)$$

where $E_{H^+}$, $E_{A^-}$, and $E_{AH}$ represent the energies of the proton, conjugate base group, and acid group. The cluster models with 66 C atoms and 22 H atoms (before substitution with −OH and −COOH) were employed for DPE calculations. The calculations for charged states were performed by applying a neutralizing uniform background charge distribution[56].

For NMR calculations, a cutoff energy of 600 eV was used and an electronic step convergence criterion was set to $10^{-9}$ eV.

## Data availability

The data to support the findings of this study are provided in the Supplementary Information and paper or from the corresponding author upon request. The DFT-structure coordinates and data that support the plots in this paper are available on Mendeley Data[57]. Source data are provided with this paper.

## Code availability

All source data are provided with this paper. All relevant data are available on Mendeley Data[57]. No specialized, home written software was used for this work.

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

## Acknowledgements

This work was financially supported by the Catalysis Center for Energy Innovation, and Energy Frontier Research Center funded by the US Department of Energy, Office of Science, Office of Basic Energy Sciences under award number DE-SC0001004. The AP-XPS was carried out in the Center for Functional Nanomaterials, Brookhaven National Laboratory, supported by the US Department of Energy, Office of Basic Energy Sciences under contract no. DE-SC0012704. Some of the XPS measurements were carried out using the Thermo Scientific K-Alpha + XPS System at the University of Delaware surface analysis facility supported by NSF (grant no. 1428149). The gas-phase absorbed ${}^{15}$N ACN NMR measurements were carried out using the 14.1T Bruker Avance III Spectrometer at the University of Delaware NMR Laboratory supported by NIH (P30 GM110758). The liquid-phase absorbed ${}^{15}$N ACN solid-state NMR measurements were supported by U.S. Department of Energy (DOE), Office of Science, Basic Energy Sciences (BES), Chemical Sciences, Geosciences & Biosciences Division, Catalysis Science program, FWP 47319, and were performed with the user proposal 60212 at Environmental Molecular Sciences Laboratory, a DOE Office of Science User Facility sponsored by the Biological and Environmental Research program under Contract No. DE-AC05-76RL01830. The authors acknowledge Dr. Wenbo Wu for assisting with the XPS measurements, Dr. Gerald Poirier, and Dr. Chinchen Kuo for assisting with the CHNS measurement and analysis.

## Author contributions

J.Z. performed all the experimental work and analysis of the XPS, experimental kinetics, gas-phase and liquid-phase absorbed ${}^{15}$N ACN ssNMR, TPDE-MS, CHNS data. P.Y. carried out all the DFT calculations. P.Y., M.C., J.Z. conducted the ML data analysis. P.K. provided guidance for catalyst synthesis and ${}^{15}$N ACN NMR data explanation. Y.C. performed the liquid-phase absorbed ${}^{15}$N ACN ssNMR, and W.S. guided the liquid-phase absorbed ${}^{15}$N ACN ssNMR. C. Q. performed the gas-phase absorbed ${}^{15}$N ACN ssNMR measurement. M.M. and A.B. performed the in situ XPS measurements. D.G.V., W.Z. and S.C. directed the project and provided guidance for the experimental and theoretical work. The manuscript was written by J.Z., P.Y., W.Z., S.C., and D.G.V. with input from all the authors.

## Competing interests

The authors declare no competing interests.
