## [Peer Review File · Nature Communications]

Tuning the reactivity of carbon surfaces with oxygen-containing functional groupsEditorial Note: This manuscript has been previously reviewed at another journal that is not operating a transparent peer review scheme. This document only contains reviewer comments and rebuttal letters for versions considered at *Nature Communications*.

REVIEWERS' COMMENTS

Reviewer #1 (Remarks to the Author):

I have reviewed the current version of the manuscript. The authors have responded to my previous comments, which were mostly focused on the nature of the active sites and the evolution of the oxygen species during the reaction. I think the authors have addressed these comments well. I understand the challenges that the authors faced to quantify the oxygen species over the spent catalysts. This is still an important fundamental question to address, maybe in a future work using a different approach. The current work nevertheless provides insights for the acidity of OH groups in graphene for dehydration of alcohols, which is interesting and could be valuable for designing carbon-based materials for heterogenous catalysis. I don't have more technical questions.

Reviewer #2 (Remarks to the Author):

Since the concerns about the results for amount of OCFGs I pointed out are addressed in the revised manuscripts, it can be considered for publication in Nature Communications.

Reviewer #3 (Remarks to the Author):

The manuscript has been revised properly, and all my concerns have been addressed. I recommend its publication on Nat Commun.

Point by point response to reviews

Reviewer No. 1

“I have reviewed the current version of the manuscript. The authors have responded to my previous comments, which were mostly focused on the nature of the active sites and the evolution of the oxygen species during the reaction. I think the authors have addressed these comments well. I understand the challenges that the authors faced to quantify the oxygen species over the spent catalysts. This is still an important fundamental question to address, maybe in a future work using a different approach. The current work nevertheless provides insights for the acidity of OH groups in graphene for dehydration of alcohols, which is interesting and could be valuable for designing carbon-based materials for heterogenous catalysis. I don't have more technical questions.”

Response: We appreciate the reviewer's comments and the feedback to improve our manuscript.

Reviewer No. 2

“Since the concerns about the results for amount of OCFGs I pointed out are addressed in the revised manuscripts, it can be considered for publication in Nature Communications.”

Response: We appreciate the reviewer's comments to improve the manuscript.

Reviewer No. 3

“The manuscript has been revised properly, and all my concerns have been addressed. I recommend its publication on Nat Commun.”

Response: We appreciate the reviewer's comments to improve the manuscript.